# Molecular dynamics simulation of entanglement spreading in generalized hydrodynamics

**Márton Mestyán[1][*] and Vincenzo Alba[2]**

**1** International School for Advanced Studies (SISSA) and INFN,
Via Bonomea 265, 34136, Trieste, Italy
**2** Delta Institute for Theoretical Physics, University of Amsterdam,
Science Park 904, 1098 XH Amsterdam, the Netherlands

[*] mestyan@fmf.uni-lj.si

## Abstract

We consider a molecular dynamics method, the so-called flea gas for computing the evolution of entanglement after inhomogeneous quantum quenches in an integrable quantum system. In such systems the evolution of local observables is described at large space-time scales by the Generalized Hydrodynamics approach, which is based on the presence of stable, ballistically propagating quasiparticles. Recently it was shown that the GHD approach can be joined with the quasiparticle picture of entanglement evolution, providing results for entanglement growth after inhomogeneous quenches. Here we apply the flea gas simulation of GHD to obtain numerical results for entanglement growth. We implement the flea gas dynamics for the gapped anisotropic Heisenberg XXZ spin chain, considering quenches from globally homogeneous and piecewise homogeneous initial states. While the flea gas method applied to the XXZ chain is not exact even in the scaling limit (in contrast to the Lieb–Liniger model), it yields a very good approximation of analytical results for entanglement growth in the cases considered. Furthermore, we obtain the *full-time* dynamics of the mutual information after quenches from inhomogeneous settings, for which no analytical results are available.

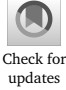
# 1   Introduction

In the last decade, the study of isolated quantum many-body systems out of equilibrium provided new insights on the deep interplay between entanglement and thermodynamics, shedding new light on the fundamental question how statistical ensembles emerge from the out-of-equilibrium dynamics after a *quantum quench* [1–6].

Integrable models offer an ideal setting for understanding generic features of the entanglement dynamics after a quantum quench [7–35]. Indeed, for integrable systems the dynamics of entanglement-related quantities can be understood within the so-called quasiparticle picture [8]. Here we focus on the out-of-equilibrium dynamics of the entanglement entropy (von Neumann entropy), which is defined as [36–39]

$$S = -\mathrm{Tr}\rho_A \ln \rho_A, \tag{1}$$

with $\rho_A$ the reduced density matrix of a macroscopic subsystem $A$ (see Fig. 1 for a one-dimensional setup). In the quasiparticle picture, pairs of entangled quasiparticles are produced after the quench. As these pairs propagate ballistically, they entangle larger regions of the system (see Fig. 1 (a)). At a given time after the quench, the von Neumann entanglement entropy is the sum of the individual contributions coming from each pair that is shared between $A$ and its complement. This picture has been explicitly verified in free-fermion models [7]. It has been shown recently that it holds true also in the presence of interactions [32].

The quasiparticle prediction for the entanglement entropy of subsystem $A$ of length $\ell$ after a quench in generic integrable systems reads as [32]

$$S(t) = \sum_{\alpha} \left[ 2t \int_{2|v_{\alpha,\lambda}|t<\ell} d\lambda |v_{\alpha,\lambda}| s_{\alpha,\lambda} + \ell \int_{2|v_{\alpha,\lambda}|t>\ell} d\lambda s_{\alpha,\lambda} \right]. \tag{2}$$

Here the index $\alpha$ labels the different types of quasiparticles present in the model, and $\lambda$ is the so-called rapidity, which distinguishes different modes of the same type of quasiparticles. The quasiparticle picture is built on two important ideas. First, at long times the entanglement

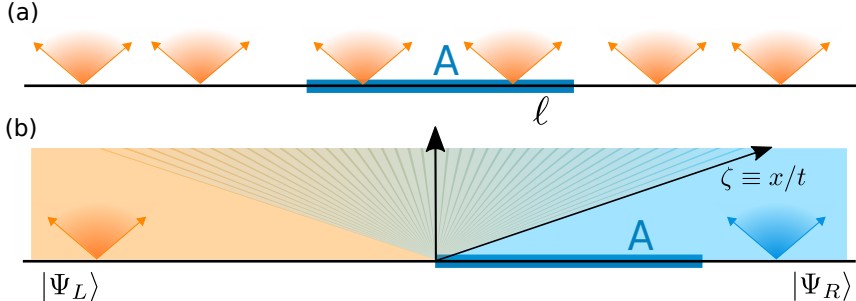

Figure 1: Dynamics of the entanglement entropy after a quench from a homogeneous (in (a)) and a piecewise homogeneous initial condition (in (b)). In both cases the entanglement dynamics is due to the ballistic propagation of pairs of entangled particles (shaded cones). In (a) the quasiparticles entanglement entropy is the thermodynamic entropy of the GGE that describes the steady state. In (b) the inhomogeneous initial state is obtained by joining two homogeneous systems (left and right) in the states $|\Psi_L\rangle$ and $|\Psi_R\rangle$. Entangled pairs are produced in the bulk of the two chains. A lightcone spreads from the interface between them. For each value of $\zeta \equiv x/t$ the system relaxes locally to a GGE. The entanglement entropy is obtained propagating the entropy of the GGEs describing the bulk of the left and right chains, i.e., for $\zeta \to \pm\infty$.

entropy between $A$ and its complement coincides with the thermodynamic entropy of the statistical ensemble that describes local steady-state properties in $A$. For generic translationally invariant steady states, this ensemble is a Generalized Gibbs Ensemble [2–6, 41–44] (GGE). Second, the velocity of the entangling quasiparticles are the group velocities of the particle-hole excitations [45] over the GGE steady state. A crucial observation is that, in contrast with free-fermion models, $v_{\alpha,\lambda}$ depends on the GGE describing the steady state after the quench (or equivalently on the pre-quench initial state), and it is "dressed" by the interactions. For generic lattice models with local interaction the velocities are finite, i.e, for any $\alpha, \lambda$, $|v_{\alpha,\lambda}| < \infty$. The function $s_{\alpha,\lambda}$, i.e., the entropy carried by quasiparticle pairs of type $\alpha$ in mode $\lambda$, is the contribution of these quasiparticles to the thermodynamic entropy of the GGE describing subsystem $A$.

The approach of Ref. [32] has been generalized in [46–48] to calculate the Rényi entropies in the steady-state, whereas calculating their full-time dynamics is a challenging open problem. Remarkably, the quasiparticle picture allows to obtain the dynamics of the logarithmic negativity [15, 49], which is a proper entanglement measure for mixed states.

Recently, there has been a growing interest in understanding the entanglement dynamics after quenches from piecewise homogeneous initial states. In the standard setup (see Fig. 1 (b)) two homogeneous chains in a different state ($L, R$ in Fig. 1) are joined together at $t = 0$. One then studies the ensuing dynamics under an integrable globally homogeneous Hamiltonian. During the time evolution a lightcone spreads from the interface between the chains. For typical initial states, in the limit of long times and large distances $x$ from the origin, the system reaches at each fixed ray $\zeta \equiv x/t$ (see Fig. 1), a Local Quasi Stationary State [50] (LQSS) which is described by a GGE. These $\zeta$-dependent GGEs can be described analytically within the Generalized Hydrodynamics (GHD) formalism [51, 52]. Furthermore, by combining the quasiparticle picture with the GHD approach [47, 53–55] it is possible, in principle, to generalize the quasiparticle picture [32] to inhomogeneous settings. However, actually calculating the full-time entanglement dynamics in this case is a demanding task. The main difficulty is that, unlike in homogeneous quenches, the trajectories of the quasiparticles are not straight

lines. Explicit results are available only in the short time limit [55] $t/\ell \to 0$ and the long time limit $t/\ell \to \infty$.

In order to overcome the above difficulties, we use a mapping between the GHD and the "molecular dynamics" of a system of classical particles called flea gas [56]. This mapping allows one to obtain the out-of-equilibrium dynamics in quantum integrable systems by performing classical simulations [57–59]. So far, the flea gas method has been employed only to integrable field theories, such as the Lieb-Liniger gas, but not to lattice models. In this paper, we discuss a generalization of the flea gas approach to the spin-1/2 anisotropic XXZ chain. An important remark is that, unlike for the Lieb-Liniger model [56], for the XXZ chain it is not straightforward to show analytically that the flea gas dynamics is fully equivalent to the GHD. In general for the XXZ chain the flea gas dynamics is expected to be different from the GHD. Our results suggest that that deviations are present, although much larger systems would be needed to ensure that the system is in the scaling limit. Still, we observe that deviations of the flea gas from the GHD are small. This results in a surprisingly good agreement between the flea gas and the GHD for the dynamics of local observables and the entanglement entropy.

Our main goal is to show that with moderate computational effort the flea gas framework allows one to compute accurate numerical predictions for the *full-time* entanglement dynamics after quenches from arbitrary inhomogeneous initial conditions, provided that the Generalized Hydrodynamics approach can be applied.

The flea gas algorithm requires to know the *bare* quasiparticle velocity and the scattering matrix between quasiparticles, which are easily obtained for a generic integrable model. The third and only nontrivial input to this method is the initial condition of the flea gas dynamics. For quenches from homogeneous initial states, the initial condition of the flea gas dynamics is given by the GGE describing the long time limit after the quench. The idea is that at comparatively short times after the quench the quasiparticles are described by the GGE density. For piecewise homogeneous setups (see, e.g., Fig. 1 (b)), the initial condition is given by the GGEs describing each homogeneous chain at $t = 0$ in the hydrodynamic timescale.

We stress that it is also necessary to know the structure of quantum correlations in the initial states, i.e., how entanglement is shared among the quasiparticles. Here we restrict ourselves to the situation in which only entangled *pairs* are present. Other situations, for instance the case of entangled "triplets", have been considered, at least in free models [26, 27].

As a benchmark of the method, we show that for quenches from homogeneous states our numerical results are in perfect agreement with (2). Moreover, for quenches from inhomogeneous initial states, in the limit $t, \ell \to \infty$ with $t/\ell \ll 1$ our results confirm the analytical predictions in Ref. [55]. Finally, to show the versatility of the method we provide results for the full-time dynamics of the entanglement entropy and of the mutual information after a quench from inhomogeneous settings, which are not easily accessible analytically [55].

The article is organized as follows. In Section 2 we introduce the XXZ chain and the quenches considered in this study. In Section 3 we discuss the Bethe ansatz treatment of generic thermodynamic ensembles. The thermodynamic Bethe ansatz framework (TBA) is introduced in Section 3.1, which is followed by the description of steady states after homogeneous quenches in Section 3.2 and a summary of the GHD approach for inhomogeneous quenches in Section 3.3. In Section 4 we introduce the flea gas method, its implementation for the XXZ chain (see Section 4.1), and the calculation of entanglement-related quantities (see Section 4.2). Our numerical results are discussed in Section 5. In Section 5.1 we benchmark the method for homogeneous quenches. In Section 5.2 we provide results for entanglement entropy after quenches from piecewise homogeneous initial states. Finally, in Section 5.3 we discuss the mutual information. Section 6 concludes the article by mentioning some interesting future directions.

## 2 Model and quenches

The flea gas method that we propose for calculating the entanglement dynamics is expected to work for generic interacting integrable models, both on the lattice and in the continuum. However, here we provide results only for a prototypical lattice model, the spin-1/2 XXZ chain, which describes a system of interacting spins on a ring, and it is defined by the Hamiltonian

$$H = \sum_{i=1}^{L} \frac{1}{2}\left(S_i^+ S_{i+1}^- + S_i^- S_{i+1}^+\right) + \Delta \sum_{i=1}^{L} S_i^z S_{i+1}^z. \tag{3}$$

Here $S_i^{+,-,z}$ are spin-1/2 operators, and $\Delta$ is the so-called anisotropy parameter. We restrict ourselves to the region with $\Delta > 1$, where the system is gapped in the thermodynamic limit, although the method can be applied for $\Delta \leq 1$ as well. We impose periodic boundary conditions by identifying sites 1 and $L + 1$. We construct our initial states by joining two homogeneous blocks that are prepared in either the translationally invariant tilted Néel state or in the translationally invariant Majumdar-Ghosh (dimer) state. The method is applicable, in principle, to any low-entangled initial state.

The translationally invariant tilted Néel state is denoted as $|N, \theta\rangle$, and it is obtained by rotating the Néel state $|\uparrow\downarrow\uparrow\dots\rangle$ around the $\hat{z}$ axis and making it translationally invariant, i.e.,

$$|N, \theta\rangle = \left(\frac{1 + \mathcal{T}}{\sqrt{2}}\right) \left\{\left[\cos(\theta/2)|\uparrow\rangle + i\sin(\theta/2)|\downarrow\rangle\right] \otimes \right.$$
$$\left. \otimes \left[\sin(\theta/2)|\uparrow\rangle - i\cos(\theta/2)|\downarrow\rangle\right]\right\}^{\otimes L/2}. \tag{4}$$

Here $\theta$ is the tilting angle and $\mathcal{T}$ is the one site translation operator to the right. The Néel state is recovered for $\theta = 0$. Similarly, the translationally invariant dimer state $|D\rangle$ is defined as

$$|D\rangle = \left(\frac{1 + \mathcal{T}}{\sqrt{2}}\right)\left(\frac{|\uparrow\downarrow\rangle - |\downarrow\uparrow\rangle}{\sqrt{2}}\right)^{\otimes L/2}. \tag{5}$$

In the homogeneous setup (Fig. 1 (a)), the chain is prepared in one of the states (4) or (5) at $t = 0$, and the system is let to evolve under (3). In the inhomogeneous case (Fig. 1 (b)) we consider quenches from the initial state $|\Psi_0\rangle = |N, \theta\rangle \otimes |D\rangle$.

## 3 Bethe ansatz description of thermodynamic macrostates in the XXZ chain

Here we introduce the thermodynamic Bethe ansatz (TBA) treatment of the XXZ chain [60], focusing on the features that are needed in the implementation of the flea gas method. First, we summarize the general TBA framework in Section 3.1. Then we report the TBA description of the steady states after the considered homogeneous quenches in Section 3.2. Finally, we summarize the generalized hydrodynamics (GHD) framework for quenches from inhomogeneous states in Section 3.3.

### 3.1 Thermodynamic Bethe Ansatz (TBA)

The XXZ chain is solved by the Bethe ansatz [60], which allows one to construct the eigenstates of (3). In the Bethe ansatz, the eigenstates are constructed with respect to the reference state with all spins up $|\uparrow\uparrow\cdots\uparrow\rangle$. Since the total magnetization $\sum_j S_j^z$ commutes with (3), the eigenstates are characterized by the total number $N$ of down spins, which is a good quantum

number of the state. We refer to $N$ as the number of particles. In this study we focus on the thermodynamic limit, i.e., the limit $L, N \to \infty$, with particle density $N/L$ fixed.

A distinctive feature of integrable models is that their eigenstates can be interpreted as a collection of well-defined, i.e., having infinite lifetime, quasiparticles. For generic integrable models the quasiparticles are labelled by a set of real parameters $\{\lambda_{\alpha,j}\}_{\alpha,j}$, which are called rapidities. For the XXZ chain at $\Delta > 1$ one has $\lambda_{\alpha,j} \in [-\pi, \pi]$. In general, there can be different species of quasiparticles. These are distinguished by the integer index $\alpha$. The total number of species depends on $\Delta$. For instance, at $\Delta \geq 1$ there is an infinite number of them. Quasiparticles with $\alpha = 1$ correspond to magnon-like excitations, whereas for $\alpha > 1$ they are bound states of $\alpha$ magnons ($\alpha$-strings [60]).

In the thermodynamic limit it is impossible to consider the individual rapidities $\lambda_{\alpha,j}$ of the quasiparticles. Instead, the standard TBA framework [60] uses the density of quasiparticles in rapidity space $\rho_{\alpha,\lambda}$, which are real functions of $\lambda$ for each species $\alpha$. One can also define the hole density $\rho_{\alpha,\lambda}^{(\mathrm{h})}$ as the density of unoccupied states in rapidity space. Another important quantity is the total density of states $\rho_{\alpha,\lambda}^{(\mathrm{t})} = \rho_{\alpha,\lambda} + \rho_{\alpha,\lambda}^{(\mathrm{h})}$. For a generic integrable model, this is a nontrivial function of $\lambda$, whereas for non-interacting systems the total density is a constant, reflecting that the rapidities are equally spaced. For the following, it is also convenient to define the filling functions $\vartheta_{\alpha,\lambda}$ and the functions $\eta_{\alpha,\lambda}$ as

$$\vartheta_{\alpha,\lambda} \equiv \frac{\rho_{\alpha,\lambda}}{\rho_{\alpha,\lambda}^{(\mathrm{t})}}, \quad \eta_{\alpha,\lambda} \equiv \frac{\rho_{\alpha,\lambda}^{(\mathrm{h})}}{\rho_{\alpha,\lambda}}. \tag{6}$$

The densities $\rho_{\alpha,\lambda}$ and $\rho_{\alpha,\lambda}^{\mathrm{h}}$ are constrained by the Bethe equations arising from the periodic boundary conditions. In the thermodynamic limit, the Bethe equations become the Bethe–Gaudin–Takahashi (BGT) equations [60]

$$\rho_{\alpha,\lambda} + \rho_{\alpha,\lambda}^{(\mathrm{h})} = a_{\alpha,\lambda} - \sum_{\beta=1}^{\infty} \int_{-\pi/2}^{\pi/2} d\mu \, T_{\alpha\beta}(\lambda - \mu) \rho_{\beta,\mu}, \tag{7}$$

where the functions $a_{\alpha,\lambda}$ are

$$a_{\alpha,\lambda} = \frac{1}{\pi} \frac{\sinh(\alpha\eta)}{\cosh(\alpha\eta) - \cos(2\lambda)}, \tag{8}$$

and $\eta \equiv \mathrm{arccosh}(\Delta)$. In (7), the scattering matrix $T_{\alpha,\beta}$ is defined as

$$T_{\alpha,\beta}(\lambda) = (1 - \delta_{\alpha\beta}) a_{|\alpha-\beta|,\lambda} + 2a_{|\alpha-\beta|+2,\lambda} + \cdots + 2a_{\alpha+\beta-2,\lambda} + a_{\alpha+\beta,\lambda}, \tag{9}$$

where $a_{\alpha,\lambda}$ is the same as in (8). The matrix $T_{\alpha,\beta}(\lambda - \mu)$ encodes all the information about the scattering between quasiparticles of type $(\alpha, \lambda)$ and $(\beta, \mu)$, and it will be crucial in the implementation of the flea gas algorithm (see section 4).

In the TBA framework, any set of particle and hole densities $\rho_{\alpha,\lambda}, \rho_{\alpha,\lambda}^{(\mathrm{h})}$ identifies a thermodynamic macrostate. All the information about thermodynamic expectation values of local and quasi local operators is encoded in the functions $\rho_{\alpha,\lambda}$. These expectation values are determined by summing over the quasiparticles species and integrating over their rapidity. For instance, for the XXZ chain the energy of a macrostate identified by a set of densities $\rho_{\alpha,\lambda}$ reads [60]

$$\frac{E}{L} = \sum_{\alpha} \int_{-\pi/2}^{\pi/2} d\lambda \, \epsilon_{\alpha,\lambda} \rho_{\alpha,\lambda}, \quad \text{with } \epsilon_{\alpha,\lambda} = -\frac{\sinh \eta \sinh(\alpha\eta)}{\cosh(\alpha\eta) - \cos(2\lambda)}. \tag{10}$$

Besides the energy, integrable models have an infinite set of quasilocal charges that commute with the Hamiltonian. In the case of the XXZ model, these charges are obtained as the derivatives of the transfer matrix. In the thermodynamic limit, a conserved quasilocal charge $\hat{Q}$ is expressed as

$$\frac{\langle \hat{Q} \rangle}{L} = \sum_\alpha \int_{-\pi/2}^{\pi/2} d\lambda \, q_{\alpha,\lambda} \rho_{\alpha,\lambda}, \tag{11}$$

where $q_{\alpha,\lambda}$ is a known function, the density of the charge.

An important quantity that we will use is the bare velocity of the quasiparticles $v_{\alpha,\lambda}^{\text{bare}}$. This is the group velocity defined from the bare quasiparticles dispersion as

$$v_{\alpha,\lambda}^{\text{bare}} \equiv \frac{\epsilon'_{\alpha,\lambda}}{p'_{\alpha,\lambda}} \qquad \text{with } \epsilon'_{\alpha,\lambda} \equiv \frac{d\epsilon_{\alpha,\lambda}}{d\lambda} \text{ and } p'_{\alpha,\lambda} = \frac{dp_{\alpha,\lambda}}{d\lambda}, \tag{12}$$

where $\epsilon_{\alpha,\lambda}$ is the bare energy of a quasiparticle defined in (10), and $p_{\alpha,\lambda}$ its bare momentum with $p'_{\alpha,\lambda} = 2\pi a_{\alpha,\lambda}$.

Interestingly, for generic integrable models the quasiparticle velocities depend on the thermodynamic macrostate [45], and they are "dressed" by the interactions. This happens because in interacting integrable models the addition or removal of a single quasiparticle causes a global rearrangement of the rapidities of the other quasiparticles. The net effect is a "dressing" of the bare quasiparticles properties, including the energies $\epsilon_{\alpha,\lambda}$, and hence the group velocities.

The correspondence between thermodynamic macrostates and microscopic eigenstates of (3) is not one-to-one. In fact, the densities $\rho_{\alpha,\lambda}$ and $\rho_{\alpha,\lambda}^{(\text{h})}$ do not uniquely determine a microscopic eigenstate. In the thermodynamic limit, the number of microscopic eigenstates that give rise to the same set of macroscopic densities diverges exponentially with the system size. The number of these thermodynamically equivalent eigenstates is given in terms of the so-called Yang–Yang entropy [60] as

$$\frac{1}{L} \ln(\# \text{ of eigenstates}) = s_{\text{YY}} = \sum_{\alpha=1}^{\infty} \int_{-\pi/2}^{\pi/2} d\lambda \, s_{\alpha,\lambda}, \tag{13}$$

with the entropy density function $s_{\alpha,\lambda}$ being

$$s_{\alpha,\lambda} = \rho_{\alpha,\lambda} \Big[ \log\big(1 + \eta_{\alpha,\lambda}\big) + \eta_{\alpha,\lambda} \log\big(1 + \eta_{\alpha,\lambda}^{-1}\big) \Big]. \tag{14}$$

The TBA formalism has been applied to describe thermal properties of the XXZ chain [60]. The corresponding thermodynamic ensemble is the Gibbs ensemble, and the Yang-Yang entropy (13) is the usual thermal entropy. However, the TBA framework can also be used to describe the thermodynamic macrostate arising after a quantum quench, i.e., the macrostate described by a generalized Gibbs ensemble (GGE) that takes into account the conservation of all the quasilocal charges. Then, the corresponding Yang-Yang entropy becomes the GGE thermodynamic entropy. Remarkably, this Yang-Yang entropy coincides with the von Neumann entanglement entropy of the post-quench steady state [32, 33, 61, 62, 65, 66], and it is one of the main ingredients to reconstruct the full-time dynamics of the entanglement entropy (as it is clear from (2)).

## 3.2 TBA treatment of the steady state after quenches from homogeneous states

Integrable models possess an extensive number of local and quasilocal conserved quantities. Their expectation value in the initial state is preserved during the dynamics. Thus, the post-quench dynamics is strongly constrained, implying that a Generalized Gibbs Ensemble, instead

of the standard Gibbs one, has to be used to describe local properties of the steady state in the long time limit. The GGE can be thought of as emerging from a generalized microcanonical principle [67]. The eigenstates entering in the microcanonical average are the ones that have the correct expectation value of the local and quasilocal conserved quantities. In the thermodynamic limit the vast majority of these eigenstates give rise to the same set of densities $\rho_{\alpha,\lambda}$. This set of densities is called the representative state. The corresponding hole densities $\rho_{\alpha,\lambda}^{(h)}$ are obtained from the BGT equations (7). The representative state encodes all information about local properties of the steady state.

Within the TBA approach there are several techniques to determine the densities $\rho_{\alpha,\lambda}$ that describe the representative state. For instance, they can be determined from the overlaps between the eigenstates of the XXZ chain with the initial state, by using the so-called Quench Action method [5]. Alternatively, they can be calculated from the knowledge of the initial values of the local and quasilocal conserved quantities [68]. The latter method allows to deal, in principle, with any translationally invariant initial state.

It is customary to describe the representative state in terms of $\eta_{\alpha,\lambda}$. These functions satisfy the so-called Y-system [69], which leads to the set of recursive equations for the functions $\eta_{\alpha,\lambda}$ (cf. (6)) as

$$\eta_{\alpha,\lambda} = \frac{\eta_{\alpha-1,\lambda+i\eta/2}\,\eta_{\alpha-1,\lambda-i\eta/2}}{1+\eta_{\alpha-2,\lambda}} - 1 \qquad \text{with } \alpha \geq 2, \tag{15}$$

with the convention that $\eta_{0,\lambda} \equiv 0$. Once $\eta_{1,\lambda}$ is known, then the functions $\eta_{\alpha,\lambda}$ for $\alpha > 1$ can be computed using (15). The corresponding particle densities $\rho_{\alpha,\lambda}$ can be computed by substituting the $\eta_{\alpha,\lambda}$ in the BGT equations (7).

Clearly, Eq. (15) implies that to determine the steady-state properties after a homogeneous global quench, one needs to calculate only $\eta_{1,\lambda}$. For both the tilted Néel state and the Majumdar-Ghosh state, which are relevant for this work, the function $\eta_{1,\lambda}$ is exactly known. For the tilted Néel state $|N, \theta\rangle$ with tilting angle $\theta$, one has [70]

$$1 + \eta_{1,\lambda} = \frac{T_1(\lambda + i\frac{\eta}{2})}{\phi(\lambda + i\frac{\eta}{2})}\,\frac{T_1(\lambda - i\frac{\eta}{2})}{\bar{\phi}(\lambda - i\frac{\eta}{2})}, \tag{16}$$

where

$$T_1(\lambda) = -\tfrac{1}{8}\cot(\lambda)\{8\cosh(\eta)\sin^2(\theta)\sin^2(\lambda) - 4\cosh(2\eta) \tag{17}$$
$$+ [\cos(2\theta) + 3][2\cos(2\lambda) - 1] + 2\sin^2(\theta)\cos(4\lambda)\}, \tag{18}$$

and

$$\phi(\lambda) = \tfrac{1}{8}\sin(2\lambda + i\eta)[2\sin^2(\theta)\cos(2\lambda - i\eta) + \cos(2\eta) + 3], \tag{19}$$
$$\bar{\phi}(\lambda) = \tfrac{1}{8}\sin(2\lambda - i\eta)[2\sin^2(\theta)\cos(2\lambda + i\eta) + \cos(2\eta) + 3]. \tag{20}$$

For the dimer state, the funcion $\eta_{1,\lambda}$ reads [68]

$$\eta_{1,\lambda} = \frac{\cos(4\lambda) - \cosh(2\eta)}{\cos^2(\lambda)(\cos(2\lambda) - \cosh(2\eta))} - 1. \tag{21}$$

### 3.3 Quenches from piecewise homogeneous initial states: Generalized Hydrodynamics

Here we consider quenches from piecewise homogeneous initial states (as described in Fig. 1 (b)). Two semi-infinite chains $L$ and $R$ are prepared in the translationally invariant states $|\Psi_L\rangle$ and $|\Psi_R\rangle$, and are suddenly joined together at $t = 0$. The ensuing dynamics is governed by the globally translational invariant Hamiltonian (3). Recently, it has been shown that a

Generalized Hydrodynamics (GHD) approach allows to study this quench [51, 52] in the long time limit at large spatial scales. Physically, during time evolution a light-cone spreads from the interface between the two chains. Outside of this lightcone, the properties of the system are the same as after the homogeneous quenches from the states $|\Psi_L\rangle$ and $|\Psi_R\rangle$ (see Fig. 1 (a)). Inside the lightcone and at late times, the expectation values of local and quasilocal observables become functions of $\zeta = x/t$. This reflects the propagation of stable quasiparticles between the two chains. This also suggests the emergence of a local quasi-stationary state for each $\zeta$. For integrable models, this corresponds to a $\zeta$-dependent GGE. Within the TBA framework, the GGE is represented by a set of TBA densities $\{\rho_{\alpha,\lambda}(\zeta)\}_{\alpha=1}^{\infty}$.

The key result of the GHD is that because of infinitely many conserved charges, the densities $\rho_{\alpha,\lambda}(\zeta)$ satisfy a simple continuity equation [51, 52] as

$$\partial_t \rho_{\alpha,\lambda}(\zeta) + \partial_x \big( v_{\alpha,\lambda}(\zeta)\rho_{\alpha,\lambda}(\zeta) \big) = 0. \tag{22}$$

In the bipartite quench the dependence of local observables on the space-time coordinates $x, t$ is only through $\zeta$. Here $v_{\alpha,\lambda}(\zeta)$ are the dressed group velocities (the same as in (2)), which are solutions of the system of integral equations [45]

$$v_{\alpha,\lambda}(\zeta) = v_{\alpha,\lambda}^{\text{bare}}(\zeta) + \sum_{\beta} \int_{-\pi/2}^{\pi/2} d\mu \frac{T_{\alpha\beta}(\lambda-\mu)}{a_{\alpha,\lambda}} \rho_{\beta,\mu}(\zeta)\big(v_{\alpha,\lambda}(\zeta) - v_{\beta,\mu}(\zeta)\big), \tag{23}$$

where $T_{\alpha\beta}$ is the scattering matrix (cf. (9) for the result for the XXZ chain) and $a_{\alpha,\lambda}$ is defined in (8). The functions $v_{\alpha,\lambda}^{\text{bare}}$ are the bare velocities defined in (12).

Physically, Eq. (23) reflects that, due to integrability, the scattering between the quasiparticles is elastic, and the only effect of the interactions is to renormalize the quasiparticles velocities. Indeed, the term $\rho_{\beta,\mu}|v_{\alpha,\lambda} - v_{\beta,\mu}|$ in (23) is the number of quasiparticles with rapidity $\mu$ and of species $\beta$ that scatter in the unit time with the quasiparticle of species $\alpha$ and rapidity $\lambda$. The ratio $T_{\alpha\beta}(\lambda-\mu)/a_{\alpha,\lambda}$ can be interpreted as an effective shift of the trajectory of the quasiparticle with label $\alpha, \lambda$ due to the scatterings. This interpretation underlies the flea gas method (cf. section 4).

The GHD approach has been successfully applied to describe transport properties in spin systems, one-dimensional integrable field theories, both classical and quantum [59, 71–74, 76–90]. Very recently, it has been shown that GHD provides a precise framework to describe experiments with trapped cold atoms [57]. A recent interesting direction is to generalize the approach to include diffusive corrections [91–95]. Finally, the GHD approach can be used to study the entanglement dynamics after quenches from piecewise-homogeneous initial states [47, 53, 55].

Unfortunately, calculating the full-time entanglement dynamics is in general a demanding task. The reason is that inside the lightcone (see Fig. 1 (b)) the trajectories of the quasiparticles are not straight lines. Explicit results are easily obtained only in some regimes. For instance, the steady-state value of the von Neumann entropy of a finite interval placed next to the interface between the two chains [53], as well as the growth rate of the entanglement entropy between two-semi-infinite systems [55], can be calculated in terms of the $\zeta = 0$ macrostate only.

## 4 Flea gas approach for out-of-equilibrium integrable systems

The flea gas approach was introduced in Ref. [96] as an effective numerical method to simulate the GHD by employing classical "molecular dynamics" techniques. The method allows to simulate the dynamics of a quantum system starting from any thermodynamic macrostate,

both homogeneous as well as inhomogeneous. So far, the approach has been implemented for the Lieb-Liniger gas but not for lattice systems such as the XXZ model.

The method was inspired by the correspondence between the continuity equation (22) and the hydrodynamic equations of a system of classical particles (hard-rod gas). Hard rods are classical one-dimensional objects undergoing elastic scattering. Here we denote their length as $d$. The hard rods dynamics is as follows. Hard rods move like free particles with bare velocity $v_{\mathrm{b}}$. When the distance between the centers of two hard rods equals $d$, they exchange their velocities. Following Ref. [96], here we adopt an alternative interpretation. One can think of hard rods as point-like objects. The scattering is then implemented as follows. When two hard rods are at the same point in space they scatter. The scattering consists of an instantaneous displacement by length $d$ of the positions of the two particles. Precisely, after assuming $d > 0$ we impose that the particle on the left (right) is shifted by $d$ to the right (left).

Let us define the density of rods with "bare" velocity $v_{\mathrm{b}}$ as $\rho(v_{\mathrm{b}})$. The number of rods with velocity between $v_{\mathrm{b}}$ and $v_{\mathrm{b}} + dv$ and in the spatial interval $dx$ is $\rho(v_b)dvdx$. During time evolution, many scatterings occur. The net effect is a renormalization of the velocity of the hard rods. Let us define this space-time dependent renormalized or "dressed" velocity as $v(v_{\mathrm{b}}; x, t)$ The dressed velocity is a function of the bare velocity $v_{\mathrm{b}}$. The density $\rho(v_{\mathrm{b}})$ obeys the continuity equation [97]

$$\partial_t \rho(v_{\mathrm{b}}; x, t) + \partial_x \big(v(v_{\mathrm{b}}; x, t)\rho(v_{\mathrm{b}}; x, t)\big) = 0. \tag{24}$$

The renormalized velocity is given by the integral equation [97]

$$v(v_{\mathrm{b}}; x, t) = v_{\mathrm{b}} + d \int dw \rho(w; x, t)\big(v(v_{\mathrm{b}}; x, t) - v(w; x, t)\big). \tag{25}$$

Equation (25) has the same structure as (23), and it admits a simple interpretation. The expression $\rho(w)|v(v_{\mathrm{b}}) - v(w)|$ is the number of scatterings per unit time between the hard rod with velocity $v$ and the ones with velocity $w$. The second term on the right hand side in (25) is the total shift that happens in the unit time to the trajectory of the hard rod with bare velocity $v$ due to the scatterings with other hard rods.

## 4.1 Flea gas for the XXZ chain and numerical implementation

We now discuss the application of the flea gas method for the XXZ chain. Before describing the method for the quenches from an inhomogeneous initial state, it is useful to consider the case of homogeneous ones. The TBA densities $\rho_{\alpha,\lambda}$ describing the steady state after the quench are stationary and homogeneous, i.e., they do not depend on $x, t$. The group velocity $v_{\alpha,\lambda}$ of the quasiparticles are obtained by solving the TBA system (23). The crucial observation is that Eq. (23) has the same structure as the equation for the hard rod gas (25). Equation (23) can be interpreted as the dressing equation for the velocities of a system of multi-species and point-like classical particles undergoing elastic scattering. Now each particle is identified by a double index $(\alpha, \lambda)$, and $T_{\alpha,\beta}(\lambda - \mu)/a_{\alpha,\lambda}$ is interpreted as a scattering length. Thus, we define $d_{\alpha,\beta}(\lambda, \mu)$ as

$$d_{\alpha,\beta}(\lambda, \mu) = \frac{T_{\alpha,\beta}(\lambda - \mu)}{a_{\alpha,\lambda}}. \tag{26}$$

Similarly to the hard rods, the particles move freely with bare velocities $v^{\mathrm{bare}}$ (now given by (12)). Scatterings occur when two particles are at the same point in space. If the particle coming from the left has labels $(\alpha, \lambda)$, and the particle coming from the right has labels $(\beta, \mu)$, then the particle coming from the left will jump $d_{\alpha,\beta}(\lambda, \mu)$, and the particle coming from the right will jump $-d_{\beta,\alpha}(\mu, \lambda)$.

A crucial remark is in order. Unlike the case of the Lieb-Liniger model [56] it is not straightforward to show that the dynamics outlined above reproduces the correct dressing for the bare velocities of the particles, i.e., Eq. (23). First, the displacement of the trajectory of a given particle is given as $\Delta x = v_{\alpha,\lambda}\Delta t$, which defines the dressed velocity $v_{\alpha,\lambda}$. The dressing of the velocity arises from the scattering with the other particles. The number of scatterings per unit time between a particle with label $(\alpha,\lambda)$ and particles with label $(\beta,\mu)$ is given as $\rho_{\beta,\mu}|v_{\alpha,\lambda} - v_{\beta,\mu}|\Delta t$.

The key issue is how to determine the direction of the jump. During the flea gas dynamics the particles move with their bare velocities. If a particle moving at $v_{\alpha,\lambda}^{\text{bare}}$ scatters with another one with velocity $v_{\beta,\mu}^{\text{bare}}$ its trajectory gets shifted by $\text{sign}(v_{\alpha,\lambda}^{\text{bare}} - v_{\beta,\mu}^{\text{bare}})d_{\alpha,\beta}(\lambda,\mu)$. For the Lieb-Liniger gas one can show that the dressed velocities are monotonic functions of the bare ones, which implies that $\text{sign}(v_{\alpha,\lambda}^{\text{bare}} - v_{\beta,\mu}^{\text{bare}}) = \text{sign}(v_{\alpha,\lambda} - v_{\beta,\mu})$. This ensures that the jumped length is $\text{sign}(v_{\alpha,\lambda} - v_{\beta,\mu})d_{\alpha,\beta}(\lambda,\mu)$. By summing over $\beta$ and integrating over $\mu$, one obtains the term on the right-hand-side in (23). This shows that the flea gas dynamics gives the correct dressing for the group velocities of the particles. On the other hand, for the XXZ chain the dressed velocities are not monotonically increasing functions of the bare ones. An important consequence is that now $\text{sign}(v_{\alpha,\lambda}^{\text{bare}} - v_{\beta,\mu}^{\text{bare}}) \neq \text{sign}(v_{\alpha,\lambda} - v_{\beta,\mu})$. This implies that for the XXZ chain one cannot conclude that the total jumped length is given as $(v_{\alpha,\lambda} - v_{\beta,\mu})d_{\alpha,\beta}(\lambda,\mu)$. To overcome this problem, our strategy here is to use the flea gas dynamics as outlined above, showing numerically that, at least in the quenches that we consider, it gives the correct dressing for the group velocities of the particles (see section 5.1).

We now discuss the details of the implementation of the flea gas method for the XXZ chain. The system is in the continuum, and it is of length $L$. Both space and time are treated as continuous variables. For a homogeneous quench, the initial state of the simulation is prepared as follows. First, we create a total number of particles $N_p$. The particles are described by the TBA densities $\rho_{\alpha,\lambda}$, which contain the full information about the post-quench GGE (see section 3.2 for the results for the quenches considered here). $N_p$ is chosen such that one has the correct value of the particle density, i.e.,

$$N_p = L \sum_\alpha \int_{-\pi/2}^{\pi/2} d\lambda \rho_{\alpha,\lambda}. \tag{27}$$

Note that $N_p$ is not the total number of down spins $N$, which is given as $N = L \sum_\alpha \int d\lambda \alpha \rho_{\alpha,\lambda}$. This simply reflects that in the simulation multi-spin bound states are treated as individual point-like particles. The particles are labeled as $1, \ldots, N_p$. Here we restrict ourselves to the situation in which only pairs of entangled quasiparticles with opposite rapidities [32] are present. For convenience, particles forming an entangled pair are labelled by consecutive integers as $(2\gamma, 2\gamma + 1)$ with $\gamma = 1, \ldots, N_p/2$. To each pair we assign a species label $\alpha$ with probability $r_\alpha$ given as

$$r_\alpha = \frac{L}{N_\text{p}} \int_{-\pi/2}^{\pi/2} d\lambda \rho_{\alpha,\lambda}. \tag{28}$$

Similarly, rapidities $\lambda_{2\gamma} = -\lambda_{2\gamma+1}$ are assigned to the pairs with probability $\rho_{\alpha,\lambda} = \rho_{\alpha,-\lambda}$. The position of each pair is random in the interval $[-L/2, L/2]$. Note that entangled particles are produced at the same point in space, implying $x_{2\gamma} = x_{2\gamma+1}$. However, to avoid spurious scatterings when the dynamics starts, we impose a tiny displacement between entangled particles. Finally, we assign to each pair their contribution to the Yang-Yang entropy, which is $s_{\alpha,\lambda}/\rho_{\alpha,\lambda}$ (cf. (14)).

During the time evolution, the particles move with the bare velocities $v_{\alpha,\lambda}^{\text{bare}}$, given as (cf (12))

$$v^{\text{bare}}_{\alpha,\lambda} = \frac{\sinh(\eta)\, a'_{\alpha,\lambda}}{2a_{\alpha,\lambda}}. \tag{29}$$

Here $a_{\alpha,\lambda}$ are defined in (8) and $a'_{\alpha,\lambda} \equiv da_{\alpha,\lambda}/d\lambda$. During the simulation only the position of the particles are updated, whereas their labels, velocities, and entropies remain the same. Particles can collide, jumping backward and forward of distance $d_{\alpha,\beta}(\lambda - \mu)$ (cf. (26)). This happens as follows. Let us denote two colliding particles as $P_1$ and $P_2$, $P_1$ being the left particle and $P_2$ the right one, respectively. Let us assume that $P_1$ and $P_2$ have labels $(\alpha, \lambda)$ and $(\alpha', \lambda')$, respectively. Thus, $P_1$ jumps to the right of distance $d_{\alpha,\alpha'}(\lambda, \lambda')$, whereas particle $P_2$ jumps to the left of distance $d_{\alpha',\alpha}(\lambda', \lambda) = -d_{\alpha,\alpha'}(\lambda, \lambda')$. It is crucial to observe that while jumping, $P_1$ and $P_2$ can scatter with other particles that are within $d_{\alpha,\alpha'}(\lambda, \lambda')$. For example, if the trajectory of $P_1$, after scattering with $P_2$, crosses that of a third particle $P_3$ with label $(\alpha'', \lambda'')$, $P_1$ scatters with $P_3$, as well. This means that, in principle, there is a "cascade" of scatterings initiating when $P_1$ and $P_2$ collide.

The complete flea gas algorithm is illustrated in Fig. 2. The first step is to identify the pair of particles $P_1$ and $P_2$ that scatter first, and the corresponding scattering time $t_{\text{coll}}$. This is performed by the routine FIND($P_1, P_2, t_{\text{coll}}$) in Fig. 2. This can be done efficiently by using standard methods in molecular dynamics simulations (see for instance Ref. [98]). Then, all the particles are evolved until $t_{\text{coll}}$, when the scattering between $P_1$ and $P_2$ occurs. This is described by the procedure COLLIDE in Fig. 2. $P_1$ and $P_2$ are instantaneously displaced by a distance $d_{1,2}$ and $d_{2,1}$ (cf. (26)). The displacement of the particles is implemented with the procedures JUMPLEFT and JUMPRIGHT, which are described in Fig. 3. Note that the two scattering particles are marked before starting the collision (see procedure MARK). This is to prevent that, while scattering with near particles, $P_1$ and $P_2$ scatter again with each other. Marked particles, instead of scattering, cross each other. After the scattering cascade starting with their first collision happened, $P_1$ and $P_2$ are unmarked.

---

| | | | |
|---|---|---|---|
| 1: **procedure** EVOLVE($t_{\max}$) | | 11: **procedure** COLLIDE($P_1, P_2$) |
| 2:      $t = 0$ | | 12:      **if** MARKED($P_1, P_2$) **then** |
| 3:      **while** $t < t_{\max}$ **do** | | 13:         $x_1 \leftrightarrow x_2$ |
| 4:         FIND($P_1, P_2, t_{\text{coll}}$) | | 14:      **else** |
| 5:         $\forall \gamma, \quad x_\gamma \rightarrow x_\gamma + v_\gamma t_{\text{coll}}$ | | 15:         MARK($P_1, P_2$) |
| 6:         $\forall \gamma$, UNMARK($P_\gamma$) | | 16:         $x_1 \leftrightarrow x_2$ |
| 7:         COLLIDE($P_1, P_2$) | | 17:         JUMPRIGHT($P_1, d_{12}$) |
| 8:         $t = t + t_{\text{coll}}$ | | 18:         JUMPLEFT($P_2, d_{21}$) |
| 9:      **end while** | | 19:      **end if** |
| 10: **end procedure** | | 20: **end procedure** |

---

Figure 2: Flea gas dynamics. The main procedure EVOLVE evolves the system up to $t_{\max}$. The routine FIND($P_1, P_2, t_{\text{coll}}$) finds the particles $P_1$ and $P_2$ that scatter first at time $t_{\text{coll}}$. The positions $x_\gamma$ of the particles are evolved up to $t_{\text{coll}}$. $v_\gamma$ are the bare velocities (cf. Eq. (29)). Then, particles $P_1$ and $P_2$ scatter. The function UNMARK removes the mark assigned to the particles when they scatter for the first time. The scattering is implemented by COLLIDE: $P_1$ and $P_2$ are displaced by a distance $d_{12} = -d_{21}$ (cf. Eq. (26)). The functions JUMPLEFT and JUMPRIGHT implementing this shift are in Fig. 3. Before scattering the particles are marked. Marked particles cross each other. Note that while $P_1$ is scattering with $P_2$, a scattering with a third particle $P_3$ can occur, initiating a scattering "cascade".

```
 1: procedure JUMPRIGHT(P₁, d)        13: procedure JUMPLEFT(P₁, d)
 2:     while d > 0 do                14:     while d < 0 do
 3:         if |x₃ − x₁| > d then     15:         if |x₁ − x₃| < |d| then
 4:             x₁ = x₁ + d           16:             x₁ = x₁ + d
 5:             d = 0                 17:             d = 0
 6:         else                      18:         else
 7:             d = d − |x₃ − x₁|     19:             d = d + |x₁ − x₃|
 8:             x₁ = x₃               20:             x₁ = x₃
 9:             COLLIDE(P₁, P₃)       21:             COLLIDE(P₃, P₁)
10:         end if                    22:         end if
11:     end while                     23:     end while
12: end procedure                     24: end procedure
```

Figure 3: Jump algorithms for the flea gas dynamics. When scattering with each other, particles $P_1$ and $P_2$ are instantaneously displaced by a distance $d$, which depends on the species and the rapidity of the particles, and it is extracted from the scattering matrix of the model (see Eq. (26)). The functions JUMPRIGHT and JUMPLEFT implement this displacement. In JUMPRIGHT and JUMPLEFT particle $P_3$ is the next particle on the right and on the left of $P_1$, respectively. If during the jump $P_1$ does not meet particle $P_3$, the position of $P_1$ is shifted by $d$. If $|x_1 - x_3| < |d|$, then particle $P_1$ scatters with $P_3$. The procedure COLLIDE is defined in Fig. 2.

## 4.2 Entanglement dynamics in flea gas simulations

The entanglement entropy at a given time is computed by counting the entangled pairs (weighted with their Yang-Yang entropy) that are shared between the subsystem of interest $A$ (cf. Fig. 1) and the rest, i.e., the number of pairs $(P_\gamma, P_{\gamma+1})$, such that $x_\gamma$ and $x_{\gamma+1}$ are in different subsystems. The result for the entanglement entropy reads as

$$S(t) = \left\langle \sum_{\substack{\text{shared pairs} \\ (\lambda, -\lambda)}} \frac{s_{\alpha,\lambda}}{\rho_{\alpha,\lambda}} \right\rangle_t. \tag{30}$$

Here the average $\langle\ \rangle_t$ is over different realizations of the flea gas dynamics up to time $t$. The sum is over the pairs that are shared between the two subsystems. Importantly, the factor $1/\rho_{\alpha,\lambda}$ takes into account that different types $(\alpha, \lambda)$ of particles appear in the sum (30) with a frequency $\ell\rho_{\alpha,\lambda}$.

## 5 Numerical results

We now provide numerical results showing the validity of the flea gas method to calculate the dynamics of the entanglement entropy after a generic quench in integrable systems. In section 5.1 we present some preliminary benchmarks of the approach. In section 5.2 we discuss the bipartite inhomogeneous quench depicted in Fig. 1 (b). Finally, in section 5.3 we discuss the mutual information between two intervals.

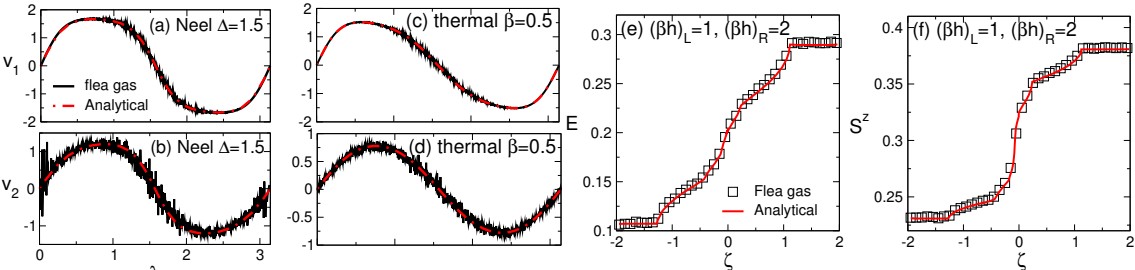

Figure 4: Flea gas versus GHD results. Panels (a) and (b) show the dressed group velocities $v_{\alpha,\lambda}$ for the first two strings plotted versus the quasiparticle rapidity $\lambda$. The black line is the flea gas result. The data are for a chain with $L = 2000$ sites and are averaged over $10^3$ realizations of the dynamics. The dash-dotted line is the solution of (23). The results shown are for the quench from the Néel state and $\Delta = 1.5$. Panels (c) and (d) show the values of $v_1$ and $v_2$ for the quench from the initial thermal density matrix (31). Panels (e) and (f) show profiles of observables in the quench of the XXZ chain with $\Delta = 2$ from the *bipartite thermal* state with $\beta_L = \beta_R = 0$, $(\beta h)_L = 1$, $(\beta h)_R = 2$, considered in Ref. [80]. We plot the local energy $E$ and the magnetization $S^z$ as a function of $\zeta \equiv x/t$. The squares represent the flea gas results averaged over $10^3$ realizations of the dynamics. The full line was obtained in [80] by solving the GHD equations (22).

## 5.1 Preliminary benchmarks

A crucial feature of the flea gas dynamics is that it gives rise to the correct dressing of the group velocities of the quasiparticles given by (23). While this can be proven for the flea gas algorithm for the Lieb-Liniger model, this is not the case for the XXZ chain. Here we provide numerical evidence that, at least for the quenches that we consider, the flea gas dynamics presented in Section 4 gives rise to the correct dressing of the group velocities.

We first consider the quench from a homogeneous chain prepared in the Néel state. Our results are presented in Fig. 4 (a) and (b). The results are for the XXZ chain with $\Delta = 1.5$. The figures show the dressed velocities of the first two strings $v_{\alpha,\lambda}$ with $\alpha = 1, 2$ plotted versus the rapidity $\lambda$. The full lines are the flea gas results. These are obtained as $\Delta x_\gamma/t$, where $\Delta x_\gamma$ is the displacement of the particles with respect to their initial position. The data are averaged over $10^4$ realizations of the flea gas dynamics. As it is clear from the Figure, there are large fluctuations in the central region around $\lambda = \pi/2$. This is because the density $\rho_{1,\lambda}$ is large at the edges of the interval, whereas it is suppressed at the center, for instance, for $\Delta = 1.5$ by a factor of $\sim 50$. This effectively reduces the statistics for the central rapidities. For $\alpha = 2$ the density $\rho_{2,\lambda}$ has a maximum around $\lambda = \pi/2$. However, it is in general much smaller than $\rho_{1,\lambda}$, again resulting in large fluctuations for the group velocities of the two strings. In Fig. 4 the dash-dotted lines are the analytical results for the dressed velocities, which are obtained by solving numerically (23). Clearly, the agreement with the flea gas results is very good.

We also considered the dressed velocities in homogeneous thermal states. Panels (c) and (d) show results for the dressed velocities in the state described by the thermal density matrix

$$\rho_0 = \frac{1}{Z} e^{-\beta H + (\beta h) S^z}, \tag{31}$$

where $\beta$ is the inverse temperature and $h$ a transverse magnetic field. The data are for $\beta = 0.5$ and $\beta h = 0.25$. The continuous lines are flea gas results for the dressed velocities of the first two strings, which perfectly match the analytical results of TBA (dash-dotted lines).

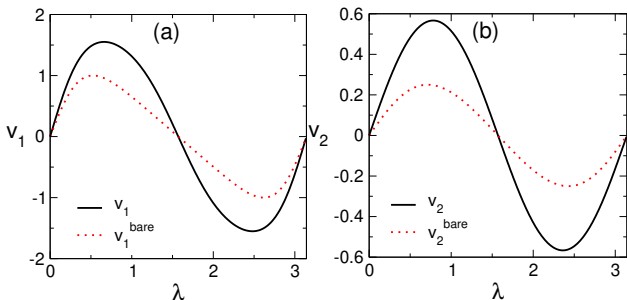

Figure 5: Comparison between bare and dressed velocities. The results are for the thermal density matrix (31) with $\beta = 0.5$, $\beta h = 0.25$ and $\Delta = 1.5$. The continuous line and the dotted line are the dressed and the bare velocities, respectively. Panel (a) and (b) show the velocities for the unbound and the two-particle bound states. Note that non monotonicity of the velocities implies that for some $\alpha, \beta, \lambda, \mu$ one has $\text{sign}(v_{\alpha,\lambda}^{\text{bare}} - v_{\beta,\mu}^{\text{bare}}) \neq \text{sign}(v_{\alpha,\lambda} - v_{\beta,\mu})$.

We now move the quenches from piecewise homogeneous states. Here we consider the initial density matrix as

$$\rho_0 = \frac{1}{Z} e^{-\beta_L H_L + (\beta_L h_L) S_L^z} \otimes e^{-\beta_R H_R + (\beta_R h_R) S_R^z}, \tag{32}$$

where quantities with the subscript $L/R$ refer to the left and right chains (see Fig. 1 (b)). The quench from (32) was investigated in Ref. [80] using GHD. Here we consider $\beta_L = 0$, $(\beta h)_L = 1$, and $\beta_R = 0$, $(\beta h)_R = 2$, with $h$ being the magnetic field ($\Delta = 2$). Due to the inhomogeneous initial condition now the dressed velocities depend on $\zeta \equiv x/t$ (see Fig. 1 (b)). To check the validity of the flea gas method, in principle, one has to check that the flea gas gives the correct result for $v_{\alpha,\lambda}(\zeta)$ for any $\zeta$. Here, instead, we consider the space-time dependence of the local energy density $E(\zeta)$ and magnetization $S^z(\zeta)$ plotted versus $\zeta = x/t$, with $t$ the time after the quench, and $x$ the distance from the origin of the lightcone. Both the quantities for $x, t \to \infty$ become functions of $\zeta$. In Figure 4 (e) and (f) the square symbols are the results of the flea gas simulation for a chain with $L = 2000$ and $t = 100$, whereas the full lines are the analytical results obtained in Ref. [80] by solving the GHD equations. The agreement between the flea gas and the GHD results is spectacular.

As a further check of the validity of the flea gas method we now discuss results for the dynamics of the von Neumann entanglement entropy after a quench from homogeneous initial states, for which analytical results (cf. Eq. (2)) are available. Our results are discussed in Fig. 7. The figure shows data for the XXZ chain with $\Delta = 2$, quenching form the Néel state (see section 2). The rescaled entropy $S/\ell$ is plotted versus $t/\ell$, with $\ell$ the subsystem size. In the simulation we considered $\ell = 100$ and a chain of length $L = 2000$. The data are obtained by averaging over $\sim 10^4$ independent realizations of the flea gas dynamics. The continuous line is the flea gas result (30) up to $t/\ell \approx 1.5$, although results for larger times can be easily obtained. The dash-dotted line is the analytical result (2) obtained in Ref. [32]. The agreement between the two is excellent.

Some remarks are in order. First, the flea gas picture is expected to capture correctly only the ballistic part of the entanglement dynamics, i.e., the leading behavior in $t/\ell$. Note that, however, subleading corrections, for instance diffusive corrections as $\mathcal{O}(\sqrt{t})$, are generically expected in the entanglement dynamics. In the flea gas framework diffusive corrections arise because of the average over the different realizations of the initial state, and are associated with the fluctuations of the particles trajectories. On the other hand, the diffusive corrections

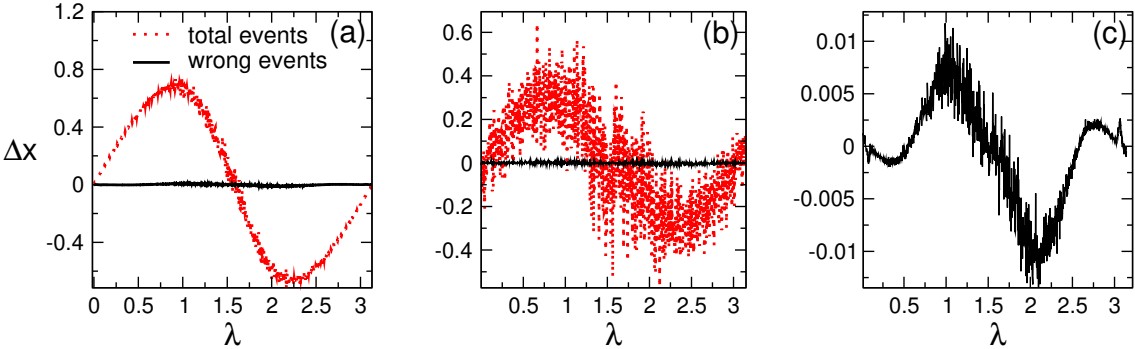

Figure 6: Effect of the "wrong" scattering events on the quasiparticles trajectories. We show the shift $\Delta x$ experienced by the quasiparticles with rapidity $\lambda$ due to scatterings with other particles. The data are for the initial thermal density matrix in (31) with $\beta = 0.5$ and $\beta h = 0.25$. (a) shows $\Delta x$ due to all the scatterings (dotted line) and only to the wrong scattering events (continuous line) for the unbound particles. In (b) we show the result for the two-particle bound states. Panel (c) shows the shift due to the wrong collisions for the unbound particles (same data as in (a)).

that are present in the flea gas are not expected to be the same as the quantum diffusive corrections of the XXZ chain. The origin of diffusion in interacting integrable models and in the flea gas have been under constant investigation in the last few years [91–95]. Finally, as it is clear from Fig. 7, subleading corrections are small. Only for very short times some deviations from (2) are present, which disappear in the scaling limit $t, \ell \to \infty$.

### 5.1.1 Effect of the wrong collisions

In section 4.1 we stressed that the fact that the group velocity of the quasiparticles are not monotonic increasing functions of the bare ones. This implies that $\mathrm{sign}(v_{\alpha,\lambda} - v_{\beta,\mu}) \neq \mathrm{sign}(v^{\mathrm{bare}}_{\alpha,\lambda} - v^{\mathrm{bare}}_{\beta,\mu})$. As a consequence two quasiparticles colliding with velocities $v^{\mathrm{bare}}_{\alpha,\lambda}$ and $v^{\mathrm{bare}}_{\alpha,\lambda}$ are shifted by the "wrong" distance $\mathrm{sign}(v^{\mathrm{bare}}_{\alpha,\lambda} - v^{\mathrm{bare}}_{\beta,\mu})d_{\alpha,\beta}(\lambda, \mu)$.

Here we investigate this effect. We consider the initial state defined by the thermal density matrix (31) with $\beta = 0.5$ and $\beta h = 0.25$. We consider the XXZ chain with $\Delta = 1.5$. The reason is that for $\Delta \to 1$ the dressing of the quasiparticles velocities is larger, which should enhance the effect of the wrong scatterings.

In Fig. 5 we compare the bare velocities and the dressed ones (dotted and continuous line, respectively). We show results only for $\alpha = 1, 2$. The effect of the dressing is clearly visible in the figure. Importantly, one consequence of the dressing is that the maximum of the velocities are shifted, as compared with the bare ones. This already implies that for some values of $\alpha, \lambda$ and $\beta, \mu$ one has that $\mathrm{sign}(v_{\alpha,\lambda} - v_{\beta,\mu}) \neq \mathrm{sign}(v^{\mathrm{bare}}_{\alpha,\lambda} - v^{\mathrm{bare}}_{\beta,\mu})$, meaning that a priori the flea gas dynamics is not fully equivalent to the GHD. On the other hand, the behavior of the bare and dressed velocities is similar as a function of $\lambda$, suggesting that the effect of the wrong scatterings should be "small".

In Fig. 6 we investigate the effect of the wrong collisions on the shift of the quasiparticles trajectories. In panel (a) we show the average total shift $\Delta x$ experienced by the quasiparticle with rapidity $\lambda$ due to wrong scatterings (continuous line) and the total number of scatterings (dotted line). The results are for the quasiparticlew with $\alpha = 1$, i.e., the unbound quasiparticles. We observe that the wrong scatterings have a small effect on $\Delta x$, which is barely visible in the figure. Similar behavior is observed for the two-particle bound states. The results are shown in panel (b). In panel (c) we show the effect of the wrong scatterings on the trajectories

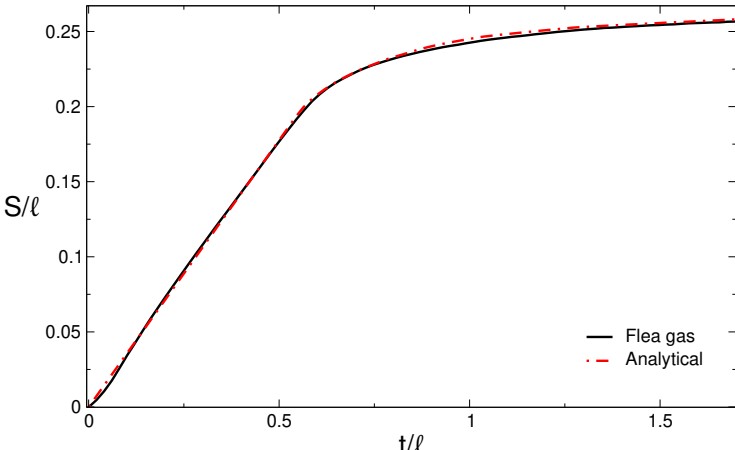

Figure 7: Dynamics of the von Neumann entanglement entropy after the quench from the Néel state in the XXZ chain with $\Delta = 2$. The entropy density $S/\ell$ is plotted versus the rescaled time $t/\ell$, with $\ell$ the subsystem size. The continuous line corresponds to the flea gas simulation for a subsystem with $\ell = 100$. The length of the chain is $L = 2000$. The results are averaged over $\sim 1000$ realization of the dynamics. The dash-dotted line is the analytical result (2).

of the quasiparticles with $\alpha = 1$. The figure shows the same data as in (a). The average shift due to the wrong scatterings is $\approx 10^{-3}$.

Finally, two observations are in order. First, although panel (c) suggests a finite contribution of the wrong scattering to $\Delta x$, larger numerical simulations would be needed to ensure that the data are in the scaling limit $N, L \rightarrow \infty$ with $N$ the number of quasiparticles. Second, in principle, it should be possible to correct the effect of the wrong scatterings by imposing that upon colliding the displacement of the quasiparticle trajectories is the correct one $\text{sign}(v_{\alpha,\lambda} - v_{\beta,\mu})$. Note that this requires knowing the dressed velocities $v_{\alpha,\lambda}$, which could be calculated during the simulation.

## 5.2 Entanglement dynamics after a quench from inhomogeneous initial conditions

Having established the validity of the flea gas method to simulate the entanglement dynamics after homogeneous quenches, we now consider the case of the inhomogeneous initial state in Fig. 1 (b). The calculation of the entanglement dynamics within the GHD framework is in general a complicated task. Explicit analytic results can be obtained only in few cases. For instance, the steady-state value of the von Neumann entanglement entropy for a finite subsystem placed next to the interface between the two chains (see Fig. 1 (b)) can be easily calculated. This corresponds to the limit $\ell/t \rightarrow 0$. In this limit, the entire subsystem is described by the GGE with $\zeta = 0$ (see section 3.3). Following Ref. [32], the density of the steady-state von Neumann entanglement entropy coincides with that of the GGE entropy with $\zeta = 0$. One has [53]

$$S = \ell \sum_{\alpha} \int_{-\pi/2}^{\pi/2} d\lambda \, s_{\alpha,\lambda}(0). \tag{33}$$

Here $s_{\alpha,\lambda}(0)$ is the Yang-Yang entropy (cf. (14)) of the GGE with $\zeta = 0$, which is obtained by using the GHD (see section 3.3). The result does not depend on which side of the system one places the interval, as long as $\ell$ is finite. Interestingly, one can show that the $\zeta = 0$ macrostate describes the entanglement growth at short times, i.e., the limit $1 \ll t \ll \ell$ as

well [53,55]. First, the entanglement entropy is expected to grow linearly at short times. Here we refer to the slope of the linear growth as the entanglement production rate [47, 53, 55]. The entanglement growth is due to the quasiparticles that cross the interface between the two chains. This suggests that the entanglement production rate is described by the $\zeta = 0$ GGE. Indeed, if subsystem $A$ is the semi-infinite chain, the entanglement production rate $S/t$ is given as [55]

$$\frac{S}{t} = \sum_\alpha \int_{-\pi/2}^{\pi/2} d\lambda \, \text{sign}(\lambda) v_{\alpha,\lambda}(0) s_{\alpha,\lambda}(0). \tag{34}$$

Here $v_{\alpha,\lambda}(0)$ is the group velocities of the particle-hole excitations around the $\zeta = 0$ GGE, which are obtained from (23). For a finite subsystem, the slope of the linear growth depends on the details of the bipartition. For simplicity we now consider the case of a finite interval of length $\ell$ placed in one of the two chains next to the interface (see Fig. 1 (b)). Clearly, the entanglement entropy gets contributions from both the edges of the subsystem. For short enough times but still in the linear regime, i.e, for large $t$ with $t/\ell \ll 1$, the contributions of the two edges decouple and can be summed independently. As in (34), one of the edges of $A$ is described by the GGE with $\zeta = 0$. On the other hand, the other one is described by the GGE with $\zeta = \pm\infty$, depending on which side subsystem $A$ is placed in. The entanglement production rate is given as

$$S = t \sum_\alpha \int_{-\pi/2}^{\pi/2} d\lambda \left[ \text{sign}(\lambda) v_{\alpha,\lambda}(0) s_{\alpha,\lambda}(0) + \left| v_{\alpha,\lambda}(\sigma\infty) \right| s_{\alpha,\lambda}(\sigma\infty) \right], \tag{35}$$

where $\sigma = \pm$ identifies the side in which subsystem $A$ is placed.

To illustrate how these features emerge in the flea gas simulations, in Fig. 8 we present numerical results for the quench from the initial state obtained by joining the Néel state and the dimer state $|N\rangle \otimes |D\rangle$ (see section 2 for the definition of these states, and section 3.2 for their TBA treatment). The results are for the XXZ chain with $\Delta = 2$. The full and dotted lines correspond to the bipartitions with interval $A$ being $[-\ell, 0]$ (in the Néel region) and $[0, \ell]$ (in the dimer region) respectively. In both cases we consider $L = 2000$ and $\ell = 100$. The results are obtained by averaging over 10000 realizations of the flea gas dynamics (see section 4), and using (30).

For the quench from $|N \otimes \text{dimer}\rangle$, we observe that at $\Delta = 2$ one has $s_{\alpha,\lambda}(+\infty) \approx s_{\alpha,\lambda}(-\infty)$ and $v_{\alpha,\lambda}(+\infty) \approx v_{\alpha,\lambda}(-\infty)$. From (35) one obtains that the entanglement production rate depends very mildly on which region the subsystem is placed. The theory predictions (35) for the entanglement production rates are not distinguishable on the scale of the figure and are reported as dash-dotted line. At intermediate $\zeta = t/\ell$, the entanglement entropy depends on all the values of $\zeta$. This happens because the entangling quasiparticles explore macrostates with different $\zeta$ as they travel in subsystem $A$ (see Fig. 1). Although it is possible, in principle, to write an analytic formula [55] for the evolution of the entanglement entropy at any time, its numerical evaluation is a demanding task. In contrast, the flea gas method allows to access easily the full-time entanglement dynamics, as it is clear from Fig. 8.

In Fig. 9 we present further checks of the validity of the flea gas method for inhomogeneous quenches. We consider the initial state obtained by joining the tilted Néel state and the dimer state, i.e., $|N, \theta\rangle \otimes |\text{dimer}\rangle$ (see section 2), where $\theta$ is the tilting angle. Panel (a) and (b) show results for $\theta = \pi/3$, whereas in (c) and (d) we consider $\theta = \pi/6$. In all the cases we choose $\ell = 100$ and total system size $L = 2000$. In (a) and (d) the subsystem is placed on the Néel side ($A = [-\ell, 0]$), whereas in (b) and (c) is in the dimer side ($A = [0, \ell]$). The fact that the production rate depends on the position of the interval is now apparent. The dash-dotted lines are the theory predictions (cf. (35)) for the entanglement production rates. In Fig. 9 (a) some deviations from (35) are visible. These, however, are due to finite-size and finite-time

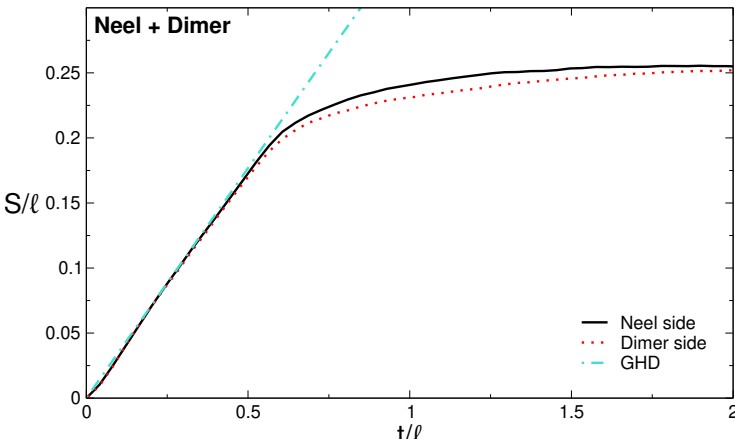

Figure 8: Dynamics of the von Neumann entanglement entropy after the quench from the initial state obtained by joining the Néel and the dimer state in the XXZ chain with $\Delta = 2$. The entropy density $S/\ell$ is plotted versus the rescaled time $t/\ell$, with $\ell$ being the subsystem size. The interval is placed next to the interface between the two chains. Here we choose $\ell = 100$, while the total chain size is $L = 2000$. The full and the dotted lines are the flea gas results for an interval placed in the Néel and dimer part, respectively. The results are obtained by averaging over $\sim 10000$ realizations of the dynamics. The dash-dotted line is the GHD prediction (35) valid in the space-time scaling limit. Notice that the asymptotic value of the entropy at $t \to \infty$ does not depend on the region where the subsystem is placed.

effects. In the inset of Fig. 9 we report results for $\ell = 500$, which are now in perfect agreement with (35). We observe that in general very large subsystems are needed to provide a robust numerical check of the GHD prediction (35).

## 5.3 Mutual information after quenches from inhomogeneous initial conditions

It is interesting to investigate the dynamics of the mutual information between two intervals. To this purpose, we now consider a tripartite system. Subsystem $A$ is made of two intervals $A_1$ and $A_2$ at a distance $d$. Here we consider only $d = 0$, although the method works also for $d > 0$. The two subsystems are embedded in an infinite system. The mutual information $I_{A_1:A_2}$ is a measure of the correlation shared between $A_1$ and $A_2$, although it is not a proper measure of the entanglement between them. $I_{A_1:A_2}$ is defined as

$$I_{A_1:A_2} \equiv S_{A_1} + S_{A_2} - S_{A_1 \cup A_2}, \tag{36}$$

where $S_{A_1}$, $S_{A_2}$, and $S_{A_1 \cup A_2}$ are the von Neumann entanglement entropies of $A_1$, $A_2$ and $A_1 \cup A_2$ with the rest of the system.

In the quasiparticle picture, the mutual information is proportional to the entangled pairs that are shared only between $A_1$ and $A_2$. On the other hand, the contribution of the quasiparticles to the mutual information is, again, the GGE thermodynamic entropy. Thus, the flea gas formula for $I_{A_1:A_2}$ is the same as (30) where the sum is restricted to the pairs of quasiparticles shared between $A_1$ and $A_2$.

The qualitative behavior of the mutual information is as follows. For two disjoint intervals at a distance $d$, the mutual information is zero at short times. At $t \sim d/t$, $I_{A_1:A_2}$ exhibits a linear increase. This corresponds to entangled pairs starting to be shared between $A_1$ and $A_2$. The growth persists up to $t \sim (d+\ell)/t$, when the mutual information starts to decrease. In systems

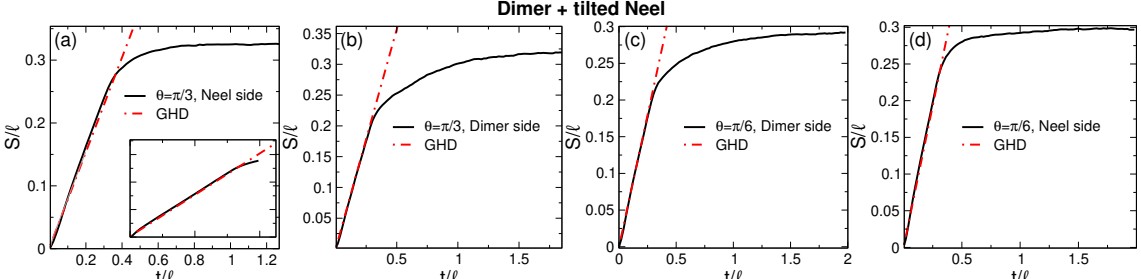

Figure 9: The same as in Fig. 8 for the quench from the state obtained by joining the tilted Néel and the dimer state. The data are for the XXZ chain at $\Delta = 2$ and for tilting angles $\theta = \pi/3$ (in (a) and (b)) and $\theta = \pi/6$ (in (c) and (d)). The curves show the flea gas results for a subsystem of size $\ell = 100$ and chain size $L = 2000$. The data are averaged over $\sim 10000$ realizations of the flea gas dynamics. In (a) and (d) the subsystem is placed on the Néel side, whereas in (b) and (c) it is in the dimer side. The dash-dotted lines are the GHD predictions in the space-time scaling limit. The inset in (a) shows results for $\ell = 500$ and chain size $L = 2000$.

with only one quasiparticle with perfect linear dispersion (as in CFT systems), the decrease is linear. In generic integrable models a much slower decrease is observed [47]. This is due to the fact that quasiparticles have a nontrivial dispersion, and slow quasiparticles entangle the two subsystems at long times. The mutual information can, in principle, be used as a tool to reveal the quasiparticle content of an integrable model. Typically, different species have different maximum velocities $v_{\alpha,M}$. This implies that if the distance between the two intervals is large enough, the mutual information exhibits a multi-peak structure in time, each peak corresponding to a different species [20, 99].

The mutual information after quenches from inhomogeneous initial states has not been investigated yet. In contrast with homogeneous global quenches [47], deriving the quasiparticle picture for the mutual information in inhomogeneous settings is a formidable task. Again, the reason is that the quasiparticles trajectories are nontrivial functions of time. We now show that the flea gas approach allows to simulate effectively the full-time dynamics of the mutual information. We restrict ourselves to the case of two adjacent intervals, although the method works for disjoint intervals as well.

We present our results in Fig. 10, focusing on the XXZ chain with $\Delta = 2$. The initial state that we consider is $|N, \theta\rangle \otimes |D\rangle$. Different panels show different values of $\theta$. The data are for two equal-length adjacent intervals $[-\ell, 0]$ and $[0, \ell]$ with $\ell = 100$. The total chain length is $L = 2000$. As expected, the mutual information is initially zero, it grows linearly at intermediate times, and it eventually decays to zero at asymptotically long times. Importantly, the initial slope of the mutual information depends only on the GGE with $\zeta = 0$ because the interface between $A_1$ and $A_2$ is at the origin. In particular, the slope of the inital growth of the mutual information coincides with the entanglement production rate for the two semi-infinite chains (see Eq. (34)). This initial growth is reported in Fig. 10 as dash-dotted line, and it perfectly describes the behavior of the flea gas results.

# 6 Conclusions

In this work we showed that the so-called flea gas method put forward in Ref. [96] provides a versatile tool for simulating the entanglement dynamics after quenches from generic ini-

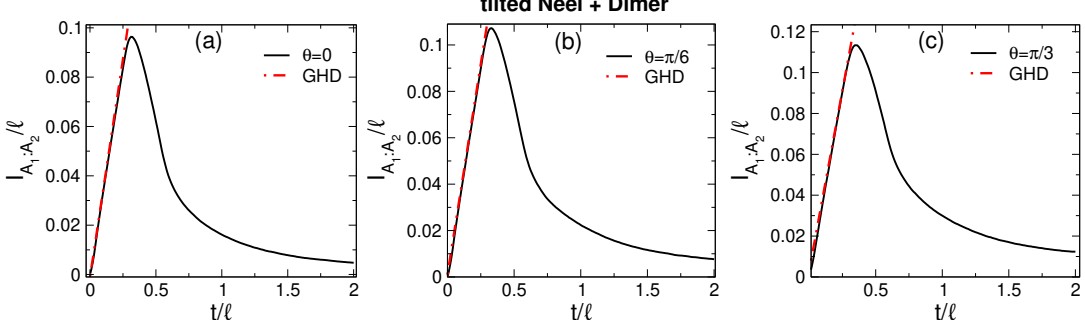

Figure 10: Dynamics of the mutual information between two intervals after the quench from the state $|\text{Néel}, \theta\rangle \otimes |\text{dimer}\rangle$ in the XXZ chain with $\Delta = 2$. Here we show the mutual information between two adjacent intervals next to the interface between the two chains. The data are for two intervals with $\ell = 100$ embedded in a chain with . The different panels are for different tilting angles $\theta$. The dash-dotted lines are the GHD predictions for the slope of the linear growth at short times. Notice that the slope depends on the macrostate with $\zeta = 0$ that describes the interface between the two chains.

tial states in integrable systems. We benchmarked the method in the Heisenberg XXZ chain, although it can be applied, in principle, to any integrable model. The method works for arbitrary initial states, both globally homogeneous as well as piecewise homogeneous. For globally homogeneous quenches the approach requires only the GGE macrostate that describes the steady-state. For piecewise homogeneous states, the key ingredients are the GGE macrostates describing the steady-state in the bulk of the two systems. Although in this case the entanglement dynamics can be obtained, in principle, by combining the quasiparticle picture with Generalized Hydrodynamics, obtaining explicit formulas [55] is a demanding task because the trajectories of the quasiparticles are nontrivial functions of time. Indeed, results can be obtained only in some limits. In contrast, in this work we showed that the flea gas approach allows to obtain easily the full-time dynamics of the entanglement entropy and of the mutual information between two intervals. Thus, the method paves the way to the study of entanglement dynamics using "molecular dynamics" simulations.

Our results open several possible new research directions. First, it would be important to investigate whether it is possible to prove analytically that for the XXZ chain the flea gas dynamics as described in section 4.1 gives the correct dressing for the group velocities.

Also, it would be useful to apply the method to more complicated setups, such as multipartite systems, or different initial states. Also, it would be important to go beyond the ballistic regime, studying corrections to the linear entanglement growth. This requires first to understand the subleading diffusive corrections in the flea gas method. Second, it requires to modify the flea gas dynamics to correctly reproduce the diffusive corrections that arise from the quantum fluctuations [92,93]. An interesting direction would be to generalize the flea gas approach to study the entanglement dynamics in the presence of defects or impurities. Finally, it would be enlightening to understand whether it is possible to treat the entanglement of operators in integrable spin chains by using the flea gas approach, generalizing the results of Ref. [100] for the Rule 54 chain.

# 7 Acknowledgements

MM acknowledges support from ERC under Consolidator grant number 771536 (NEMO). VA acknowledges support from the D-ITP consortium, a program of the NWO. VA also acknowledges support from the European Research Council under ERC Advanced grant 743032 DY-NAMINT.

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
