# Peer review of "Molecular dynamics simulation of entanglement spreading in generalized hydrodynamics"

_SciPost Physics, doi:SciPost Phys. 8, 055 (2020)_

## Round 2 · Referee Report · Anonymous (Referee 1) · 2019-8-27

Strengths

  1. timely subject exploring an effective approach to the non-equilibrium dynamics of quantum integrable systems
  2. displays application of the molecular dynamics methods to the quantum lattice models
  3. raises an interesting question on analytical derivation of the flea gas dynamics for the XXZ spin chain

Weaknesses

  1. lack of physical explanation why flea gas method is applicable to the XXZ spin chain, see the report for the details.

Report

The authors apply the flea gas method, introduced and studied previously for the continuum models, to the dynamics of the XXZ spin chain. They focus on the entanglement spreading and compare the results of the simulations to the GHD at the Euler scale. The results suggest that the flea gas method is applicable in the lattice settings.

The paper is well written and the exposition of the subject, method and results is very good. Still, there are few points that I would like the authors to address.

1) page 3: The statement that the initial state is described by the GGE describing the long time limit is confusing. Please explain.

2) page 9: The bare velocity is used to uniquely distinguish different hard-rods. This requires however a Pauli-like principle in the system. Please comment on this.

3) page 9 and other places: the authors refer to $\rho(v(v_b) - v(w))$ or similar expressions as giving the number of scattering events per unit time. The units however does not match. Please correct.

4) page 10: following the discussion the flea gas dynamics seems inappropriate for the XXZ spin chain because the dressing can change the ordering of two velocities. Still the results suggest that the resulting dynamics is correct. I would like the authors to expand on this briefly answering or commenting on the following questions:

  • how strong is the violation of the monotonicity?

  • how does it change the dynamics?

  • why is this effect negligible in the situations studied?

  • are there situations in which this effect would be stronger?

5) page 12: what values does index $\alpha$ take in the flea gas algorithm, figure 2?

6) page 14: it is unclear if in figs. 4 e) & f) plotted are results of a single simulation or an average of many simulations. Please correct.

7) page 15: please explain what happens to the term with an explicit t dependence in (2) when writing it in the $l/t \rightarrow 0$ limit as in (33).

I would like the authors to address these questions before suggesting the publication of this work.

Requested changes

Beside the issues mentioned in the report there are few smaller things:

8) page 2: the authors write "In (2), $v_{\alpha, \lambda}$ are these velocities". I find this sentence confusing/unnecessary.

9) page 4: is there a particular reason why the initial state of the form $|N, \theta\rangle \otimes |D\rangle$ is interesting?

10) page 8: a typo in the first paragraph "... to study the this quench [50,51]."

11) page 10: in the first paragraph first $\nu$ then $\lambda$ are used to refer to apparently the same particle

12) page 11: von Neumann entropy -> von Neumann entanglement entropy

---

## Round 3 · Referee Report · Anonymous (Referee 1) · 2020-2-24

Report

I thank the authors for addressing my concerns and recommend the paper for the publication.

---

## Round 3 · Author Response

We thank the referee for her/his careful reading of the manuscript. Here are our answer to her/his questions and comments.

1) page 3: The statement that the initial state is described by the GGE describing the long time limit is confusing. Please explain.

RESPONSE: The quasiparticle picture holds after local relaxation at the microscopic scale has happened, and when local observables of the system are already described by a GGE. This happens at a time scale that is comparatively small with respect to the long time limit in which the quasiparticle picture holds. Therefore the GGE can be viewed as an input to the quasiparticle picture. We added a sentence at page 3 in the fourth paragraph to clarify this.

2) page 9: The bare velocity is used to uniquely distinguish different hard-rods. This requires however a Pauli-like principle in the system. Please comment on this.

RESPONSE: In the classical hard rod gas, velocity has a continuous distribution and the average number of particles in a volume $[v, v+d v]$ is given by $\rho(v) d v$. In a realization of the hard rod gas, where velocities of particles are drawn from $\rho(v)$, each particle will have a unique velocity since the probability of two velocities coinciding \emph{exactly} is zero. On a technical level, this fact can be used to distinguish among particles in the simulation. However, this fact has no physical relevance (and particularly no connection with the Pauli principle), so in the second paragraph at page 9 we removed the statement that the velocity can be used to distinguish different hard rods.

3) page 9 and other places: the authors refer to ρ(v(vb)−v(w)) or similar expressions as giving the number of scattering events per unit time. The units however does not match. Please correct.

RESPONSE: On page 9, the density $\rho(w)$ in expression $\rho(w)(v(v_b) - v(w))$ is a density in the space of velocities of hard rods, and has dimension 1/([L][v]). This is now mentioned explicitly in the second paragraph at page 9. In the case of the flea gas simulating the XXZ model, $\rho(\lambda)$ is the density in the space of rapidities, and has dimension $1/[L]$.

4) page 10: following the discussion the flea gas dynamics seems inappropriate for the XXZ spin chain because the dressing can change the ordering of two velocities. Still the results suggest that the resulting dynamics is correct. I would like the authors to expand on this briefly answering or commenting on the following questions:

  • how strong is the violation of the monotonicity?

  • how does it change the dynamics?

  • why is this effect negligible in the situations studied?

  • are there situations in which this effect would be stronger?

RESPONSE: This is a very important point and we thank the referee for raising it. Following the referee suggestion we decided to investigate this issue. As stated in the paper for the XXZ chain the flea gas dynamics in not expected to be the same as the GHD. As correctly summarized by the referee, the reason is that the group velocities for a generic XXZ state are not monotonic functions of the rapidity. This implies that during the flea gas dynamics some of the quasiparticles will undergo wrong'' jumps, i.e., jumps in the wrong direction. On the other hand, our numerical results for both local observables and for the entropy suggest that the discrepancy between the flea gas and the GHD are absent or small. We investigate this issue in the newly added section 5.1.1. We consider the flea gas for the thermal density matrix in formula (31). We choose the anisotropy Delta=1.5. The reason is that we observe that the dressing of the bare velocities is larger for $\Delta \to 1$, which should enhance the effect of thewrong'' collisions. First we compare in Fig. 5 the bare and the dressed velocities. Although the bare and dressed velocities have the same qualitative behavior as functions of the rapidity, the maxima and minima are shifted by the dressing. This implies that for instance near a maximum of the bare velocities the dressed velocities can be monotonically increasing or decreasing. This implies that for two rapidities $\lambda,\mu$ one has

$sign(v^b_\lambda -v^b_\mu) \ne sign(v_\lambda-v_\mu)$

To investigate how strong this effect is, in Fig. 6 we present flea gas data for the average spatial shift experienced by the quasiparticles, due to the collisions, which is used to calculate the dressing of the velocity. Since we can calculate the dressed velocity using GHD, we can also distinguish
the shifts due to the wrong collisions. In Fig. 6 we report the total average shift and the one due to the wrong collisions for the unbound particles and for the bound states of two particles. The Figure shows that the effect of the wrong collisions is larger near the maxima and minima of the velocities, as expected. However, the effect is in general quite small, explaining the generic good agreement between the flea gas simulations and the GHD results that we observe in the paper. Finally, we should mention that this observation could be in principle exploited to improve the flea gas method. The idea would be to estimate the dressed velocities during the simulation, comparing it to the bare ones, which are an input of the simulation, and using this to correct the sign of the shift of the quasiparticles during the wrong collisions.

5) page 12: what values does index α take in the flea gas algorithm, figure 2?

RESPONSE: In line 6 of Figure 2, $P_\alpha$ should run over all the pairs, i.e., all the previous markings have to be removed. We have modified the line accordingly to clarify this (we changed the index from $\alpha$ to $\gamma$, and added $\forall$).

6) The plots in fig 4 e), f) show the average over 10^3 simulations. We have added this fact to the figure caption.

RESPONSE: The plots in fig 4 e), f) show the average over 10^3 simulations. We have added this fact to the figure caption.

7) page 15: please explain what happens to the term with an explicit t dependence in (2) when writing it in the l/t→0 limit as in (33).

RESPONSE: In the $\ell/t \rightarrow 0$ (or $1/t \rightarrow 0$) limit, the first term in (2) tends to zero. The reason for this is the following: since the velocities $v_{\alpha,\lambda}$ as a function of $\lambda$ vanish with finite first-order derivative (the argument can be generalizes to the case in which the first derivative vanishes), the length of the domains of integration in $\lambda$ scale as $O(1/t)$. Consequently the velocities $v_{\alpha,\lambda}$ inside these domains also scale as $O(1/t)$. Together with the factor of $2 t$ before the summation, the scaling of the term is $O(t^{1-1-1})=O(t^{-1})$, therefore it goes to zero in the $1/t \rightarrow 0$ limit. (The factor $s_{\alpha,\lambda}$ in the integrand is bounded from above so it does not affect the limit value.)

Beside the issues mentioned in the report there are few smaller things:

8) page 2: the authors write "In (2), vα,λ are these velocities". I find this sentence confusing/unnecessary.

9) page 4: is there a particular reason why the initial state of the form |N,θ⟩⊗|D⟩ is interesting?

10) page 8: a typo in the first paragraph "... to study the this quench [50,51]."

11) page 10: in the first paragraph first ν then λ are used to refer to apparently the same particle

12) page 11: von Neumann entropy -> von Neumann entanglement entropy

RESPONSE: We thank the referee for these minor corrections. We have modified the manuscript accordingly.

---

## Round 3 · List of Changes

• The discussion of the approximative nature of the flea gas in the XXZ model has been extended. A new figure (Figure 6) has been added.
  • The abstract has been rewritten to reflect this approximative nature.
  • Typos corrected and several minor corrections (see the response above).

---

## Editorial Decision

published